# Recursive Transformer: Boosting Reasoning Ability with State Stack

**Kechi Zhang**[1,2]**, Ge Li**[1,2]*,**Jia Li**[3]**, Huangzhao Zhang,**
**Yihong Dong**[1,2]**, Jia Li**[4]**, Jingjing Xu**[5]**, Zhi Jin**[1,2,3]*
[1]Key Lab of High Confidence Software Technology (PKU), Ministry of Education
[2]School of Computer Science, Peking University, China
[3]School of Computer Science, Wuhan University, China
[4]College of AI, Tsinghua University
[5]ByteDance
{zhangkechi,lige,zhijin}@pku.edu.cn, zhijin@whu.edu.cn

## Abstract

The Transformer architecture has emerged as a landmark advancement within the broad field of artificial intelligence, effectively catalyzing the advent of large language models (LLMs). However, despite its remarkable capabilities and the substantial progress it has facilitated, the Transformer architecture still has some limitations. One such intrinsic limitation is its inability to effectively recognize regular expressions or deterministic context-free grammars. Standard Transformers lack an explicit mechanism for recursion and structured state transitions, which can hinder systematic generalization on nested and hierarchical patterns. Drawing inspiration from pushdown automata, which efficiently resolve deterministic context-free grammars using stacks, we equip layers with a differentiable stack and propose STACKTRANS with recursion to address the aforementioned issue within LLMs. Unlike previous approaches that modify the attention computation, STACK-TRANS explicitly incorporates hidden state stacks between Transformer layers. This design maintains compatibility with existing frameworks like flash-attention. Specifically, our design features stack operations – such as pushing and popping hidden states – that are differentiable and can be learned in an end-to-end manner. Our comprehensive evaluation spans benchmarks for both Chomsky hierarchy and large-scale natural languages. Across these diverse tasks, STACKTRANS consistently outperforms standard Transformer models and other baselines. We have successfully scaled STACKTRANS up from 360M to 7B parameters. In particular, our from-scratch pretrained model STACKTRANS-360M outperforms several larger open-source LLMs with 2–3× more parameters, showcasing its superior efficiency and reasoning capability.

## 1 Introduction

In the era of large language models (LLMs) [GPT-4, 2023], the Transformer architecture has emerged as the nearly universal backbone, achieving remarkable success across various domains and beyond [Bi et al., 2024; Bai et al., 2023; Zhang et al., 2024a]. However, recent empirical studies [Delétang et al., 2022; Hahn, 2020] have demonstrated that Transformers struggle with tasks involving Chomsky hierarchy [Chomsky, 1956], such as regular expressions (REs) and deterministic context-free grammars (DCFs) with hierarchical/recursive structure. In formal language theory, deterministic context-free languages are a proper subset of context-free languages. For example, in an RE match-

---

*Ge Li and Zhi Jin are the corresponding authors.

39th Conference on Neural Information Processing Systems (NeurIPS 2025).

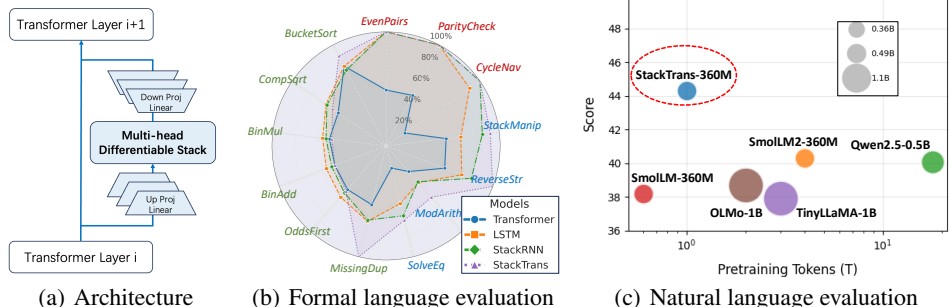

| (a) Architecture | (b) Formal language evaluation | (c) Natural language evaluation |

Figure 1: STACKTRANS architecture and its overall performance (best view in color). (a) The hidden state stack can be integrated between Transformer layers to model the Chomsky hierarchy. (b) Radar chart for formal language tasks, where task difficulty is indicated by color (from red to blue to green, with green representing the most difficult tasks). (c) Results on natural language tasks, where the $x$-axis represents the pretraining corpus size, the $y$-axis represents the average performance across multiple benchmarks (see Table 2), and the dot size indicates the model size.

ing task, each training example consists of a string within a specified length range, paired with a matching-or-not label. While Transformers can perform well within the length boundaries of the training set, they often fail to maintain consistent performance when evaluated on input strings that are longer or shorter than those in the training data. One theoretical explanation is that standard Transformers lack suitable inductive biases—most notably, an explicit mechanism for recursion and reusable structured state transitions [Mitchell, 1980; Battaglia et al., 2018; Sartran et al., 2022]. Without inherent assumptions to guide learning, Transformers struggle to effectively capture the underlying grammar in the training set. Consequently, they cannot generalize beyond the training data and perform poorly on inputs of lengths that differ from those seen during training. On the other hand, natural languages are generally considered to belong to a class of languages that extends beyond DCF [Gazdar and Pullum, 1982; Chomsky, 1956; Shieber, 1985]. This inherent limitation of Transformers may possibly hinder their application in real-world natural language modeling. Such deficiencies may also have the risk to block LLMs to achieve more advanced forms of intelligence.

In the realm of formal language theory and computational linguistics [Chomsky, 1956; Savage, 1998; Sipser, 1996], it is widely recognized that automata augmented with stacks correspond to different levels of the Chomsky hierarchy grammars [1]. Given the success of stack-equipped automata in handling rather complex grammars, it is a natural progression to incorporate the data structure of stack into the Transformer architecture. Recently, researchers have proposed the stacked attention mechanism [DuSell and Chiang, 2024; Li et al., 2024] and have examined its viability upon multiple relatively small benchmarks, but these approaches primarily restructure attention rather than introducing an explicit, reusable recursive state mechanism.

Drawing inspiration from pushdown automata and prior research, we introduce hidden state stacks into the Transformer architecture, proposing a novel method, STACKTRANS. We use the term "stack" because the core update mechanism of our module is fundamentally based on differentiable push and pop operations. The proposed mechanism improves hierarchical generalization and recursive reasoning in Transformers. Unlike the stacked attention mechanisms [DuSell and Chiang, 2024; Li et al., 2024], which replace the standard attention, STACKTRANS incorporates differentiable stacks between the Transformer layers, meanwhile preserving the integrity of the Transformer layers (Figure 1(a)). This integration allows us to embed the assumptions of the Chomsky hierarchy into the model, enabling STACKTRANS to inherently model and learn REs and DCFs. Moreover, this design maintains compatibility with existing frameworks like flash-attention [Dao et al., 2022; Dao, 2024], enabling seamless integration into efficient LLM training pipelines. Specifically, the stack stores hidden states generated by the preceding layer and updates through operations such as push

---

[1]The Chomsky grammars of type 3 (REs) and type 2 (DCFs) can be recognized by pushdown automata (automata equipped with a stack) [Chomsky, 1956; Chomsky and Schützenberger, 1959]. With proper enhancement, pushdown automata can even resolve type 1 (context-sensitive, CS) and type 0 (unrestricted) grammars – 2-stack automata (automata equipped with two independent stacks) are equivalent to Turing machines, and can recognize type 0 grammars [Yau, 1969].

and pop at each decoding time step. To enable end-to-end training of STACKTRANS, we design soft stack operations, thereby making the hidden state stack differentiable. STACKTRANS also adopts a multi-head stack to improve its representation capability. Additionally, we find that the standard stack reading operation (which only returns the top element in the stack) may cause unstable training. Therefore, we propose the global reading operation through a learnable query-over-stack attention, stabilizing the training process and enriching the expressiveness of STACKTRANS.

We conduct comprehensive experiments on multiple benchmarks spanning both formal languages [Delétang et al., 2022] and natural languages [Groeneveld et al., 2024]. Specifically, in RE and DCF tasks, STACKTRANS outperforms the standard Transformer by at least 30%, achieving nearly 100% test accuracy on all regular language tasks and demonstrated excellent performance on the majority of deterministic context-free grammar tasks, as shown in Figure 1(b). Notably, during natural language evaluation, STACKTRANS also demonstrates substantial improvements on tasks such as common sense reasoning and question answering. Furthermore, we have successfully scaled STACKTRANS up from 360M to 7B parameters. In particular, our open-sourced STACKTRANS-360M, which is pretrained on a corpus of approximately 1T tokens, performs better than or comparably to state-of-the-art LLMs with 2–3$\times$ more parameters, as shown in Figure 1(c).

The contribution of this paper can be summarized as below:

❶ We introduce STACKTRANS, which incorporates hidden state stacks between Transformer layers, providing an explicit mechanism to model recursive structure. This integration enables STACK-TRANS to inherently learn grammars from the Chomsky hierarchy, including REs and DCFs.

❷ We design the hidden state stack to be differentiable, which employs soft push, pop, and no-op operations, a multi-head stack mechanism, and a global stack reading operation. These innovations ensure stable and end-to-end training of STACKTRANS.

❸ We conduct extensive experiments on multiple formal language benchmarks, demonstrating STACKTRANS's effectiveness in learning Chomsky hierarchy grammars such as REs and DCFs. Furthermore, we have successfully scaled STACKTRANS up from 360M to 7B parameters for general language modeling. Evaluations on large-scale natural language benchmarks show that STACKTRANS-360M outperforms baselines with even $2 - 3\times$ more parameters.

## 2  Background

The foundational success of LLMs can be attributed to the development of the Transformer architecture [Vaswani et al., 2017] and its numerous variations, which serve as the backbone for LLMs [Radford et al., 2018; GPT-4, 2023; GPT-4o, 2024]. Despite their widespread adoption, the Transformer architecture has inherent expressivity limitations [Hahn, 2020]. Recent studies have shown that standard Transformers struggle with recursive and hierarchical patterns across both synthetic and real-world tasks [Joulin and Mikolov, 2015; Grefenstette et al., 2015; Sartran et al., 2022], and need to equip neural networks with external data structures. These limitations pose a potential risk to the natural language modeling capabilities of Transformers, as suggested by discussions surrounding the classification of natural languages within the Chomsky hierarchy [Gazdar and Pullum, 1982; Chomsky, 1956; Shieber, 1985]. For more detailed related work, please refer to §A. Based on the same principle, STACKTRANS introduces differentiable hidden state stacks in a modular and scalable manner. Without altering the attention mechanism, STACKTRANS integrates stack operations into layer-wise hidden state updates. This design maintains compatibility with existing frameworks like flash-attention [Dao et al., 2022; Dao, 2024] and supports architectures ranging from 0.36B to 7B parameters. By learning stack operations explicitly, STACKTRANS is capable of addressing broader linguistic and algorithmic challenges.

## 3  Method

As introduced previously, the standard Transformer architecture [Vaswani et al., 2017] struggles to learn the Chomsky hierarchy due to its lack of inductive biases. To address this limitation, we propose STACKTRANS, which incorporates hidden state stacks into the Transformer architecture. These stacks introduce the assumptions of the Chomsky hierarchy, enhancing the model's ability to capture hierarchical structures. In STACKTRANS, the hidden state stacks augment the information flow by

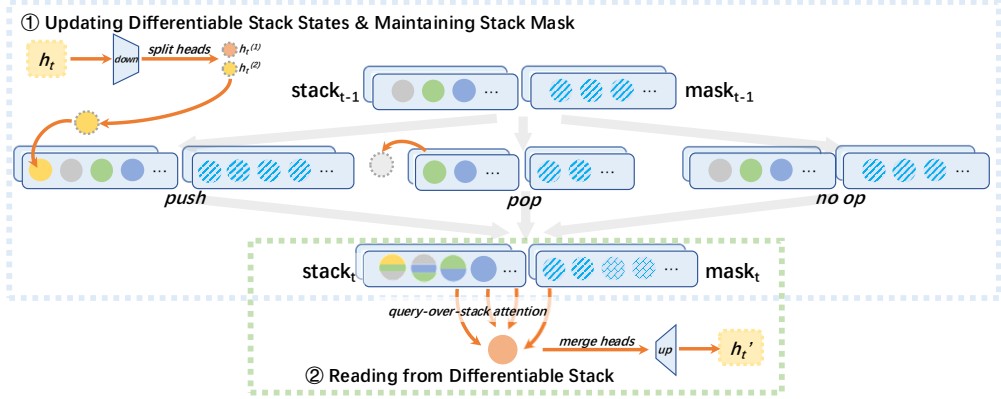

Figure 2: Illustration of the multi-head differentiable stack. Our designed differentiable stack includes the updating mechanism based on three action probabilities (*i.e.*, push, pop, and no-op), the stack mask maintenance, as well as a gated global reading mechanism. To improve both memory efficiency and representational flexibility, we also add multi-head and low-rank mechanisms.

routing token-level hidden states through learnable stack operations, such as stack updating and reading. Specifically, the stacks are integrated between standard Transformer layers, where they store hidden states and perform updates and readings via soft operations (see §3.1 for details). To further improve the expressiveness of STACKTRANS, we introduce a multi-head stack mechanism (please refer to §3.2). This design enhances the ability of STACKTRANS to capture diverse patterns with low-rank representations. Finally, to ensure robust training and avoid issues such as stack operation collapsing, we introduce stack regularization techniques. We also implement stack truncation to facilitate batching and parallel training (§3.3). These innovations and designs collectively enhance the effectiveness and stability of STACKTRANS in learning Chomsky hierarchy grammars.

## 3.1 Hidden State Stack

In computational linguistics, it is well established that the Chomsky hierarchy grammars can be resolved by different classes of automata, with pushdown automata being the minimal computational model for DCFs [Chomsky and Schützenberger, 1959]. Drawing inspiration from this, we incorporate stacks into the Transformer architecture to address its limitations in learning the Chomsky hierarchy. Recall that a standard stack is a last-in-first-out storage structure that allows the top element to be read or updated (*i.e.*, operations like peek, push, and pop). Given a hidden state sequence $h_1, \cdots, h_l$ (for now, we do not consider any computational dependencies among $h_t$'s), at each time-step $t$, STACKTRANS is designed to either push the current hidden state $h_t$ into the stack or pop the top element from the stack, and then peek the current top element (at this stage, we ignore how these operations are determined or executed by the model). This procedure results in a new stack-operated sequence, which is a permutation of $h_1, \cdots, h_l$. Ideally, if the stack operations are correct, it is highly plausible that the model can effectively learn the target Chomsky hierarchy grammar.

Since the hidden state stack will be incorporated into STACKTRANS, it must be differentiable to enable end-to-end training. However, standard stack operations such as push and pop are discrete, which disrupts gradient back-propagation and hinders the training process. To address this challenge, we introduce a soft operation mechanism, following earlier explorations [Grefenstette et al., 2015; Joulin and Mikolov, 2015]. In this mechanism, the results of the stack operations are continuously interpolated based on some trainable parameters. This design not only makes the operations differentiable but also allows the model to learn the stack operations through these trainable parameters.

**Soft update** The stack at time-step $t$ can be formalized as a list of vectors, where $\text{St}_t[i]$ refers to the $i$-th element from top to bottom in the current stack $\text{St}_t$. We define three candidate operations for STACKTRANS determined by the current hidden state $h_t$ (here, $\text{St}_t[i]$ and $h_t$ both belong to $\mathbb{R}^d$, meaning they share the same width $d$): ❶ "push" shifts every element down by one position and puts $h_t$ at the top; ❷ "pop" removes the top element and moves every remaining element up by one position; and ❸ "no-op" does not alter $\text{St}_t$ at all. Instead of discretely selecting one of these

operations, the soft update mechanism computes a distribution over the candidate operations. This is achieved through a learned linear projection $A \in \mathbb{R}^{3 \times d}$ followed by a softmax function:

$$a_t = [a_t^{\text{push}}; a_t^{\text{pop}}; a_t^{\text{noop}}] = \text{Softmax}(Ah_t) \tag{1}$$

where each scalar in $a_t$ represents the probability corresponding to one operation. We then combine the results of each operation based on their respective probabilities to update the stack as follows:

$$\text{St}_{t+1}[i] = \begin{cases} a_t^{\text{push}} \cdot h_t & + & a_t^{\text{pop}} \cdot \text{St}_t[1] & + & a_t^{\text{noop}} \cdot \text{St}_t[0], & \text{if } i = 0 \\ a_t^{\text{push}} \cdot \text{St}_t[i-1] & + & a_t^{\text{pop}} \cdot \vec{0} & + & a_t^{\text{noop}} \cdot \text{St}_t[i], & \text{if } i = S-1 \\ a_t^{\text{push}} \cdot \text{St}_t[i-1] & + & a_t^{\text{pop}} \cdot \text{St}_t[i+1] & + & a_t^{\text{noop}} \cdot \text{St}_t[i], & \text{otherwise} \end{cases} \tag{2}$$

where $S$ denotes the size of the stack. The first and the second rows in Equation 2 correspond to the top element ($\text{St}_{t+1}[0]$) and the bottom element ($\text{St}_{t+1}[S-1]$) respectively, while the last row pertains to the intermediate elements. It is important to note that a zero vector is always maintained at the bottom of the stack, as indicated in the "pop" term of the second row in Equation 2.

The soft update mechanism is fully differentiable, enabling end-to-end parameter tuning. Meanwhile, the dynamic of the information flow aligns with that of a standard stack. When $a_t^{\text{push}}$ dominates in $a_t$, all elements in the stack tend to shift downward as more information from $h_t$ flows into the top element; conversely, when the pop operation dominates, the elements in the stack shift upward as the information of the top element is mostly removed.

**Stack mask** To implement our proposed hidden state stack using a list, the overall available stack size $S$ must be sufficiently large. Assuming $S$ is large enough, the tail of the stack is always padded with zero vectors. This padding ensures that stack operations comply with logical constraints and prevents invalid behaviors. However, this process is inherently discrete. To address this, we propose maintaining a differentiable stack mask for STACKTRANS. Specifically, the $i$-th element in the mask $\text{M}_t \in \mathbb{R}^S$ suggests how likely the corresponding element in the stack is active. The mask is updated with dynamics similar to those described in Equation 2:

$$\text{M}_{t+1}[i] = \begin{cases} a_t^{\text{push}} \cdot 1 & + & a_t^{\text{pop}} \cdot \text{M}_t[1] & + & a_t^{\text{noop}} \cdot \text{M}_t[0], & \text{if } i = 0 \\ a_t^{\text{push}} \cdot \text{M}_t[i-1] & + & a_t^{\text{pop}} \cdot 0 & + & a_t^{\text{noop}} \cdot \text{M}_t[i], & \text{if } i = S-1 \\ a_t^{\text{push}} \cdot \text{M}_t[i-1] & + & a_t^{\text{pop}} \cdot \text{M}_t[i+1] & + & a_t^{\text{noop}} \cdot \text{M}_t[i], & \text{otherwise} \end{cases} \tag{3}$$

$\text{M}_t$ serves as an activation controller in STACKTRANS– if $a_t^{\text{push}}$ dominates, one more stack element is further activated, otherwise $a_t^{\text{pop}}$ dominates in $a_t$, the last activated element is more likely to be deactivated. When accessing the stack, we pad the stack by element-wise production of $\text{St}_t$ and $\text{M}_t$.

**Global read** The standard stack simply peeks and returns the top element during reading. Although peeking is quite straightforward, we notice that it may cause unstable training during our initial experiments. Furthermore, limiting access to only the top element restricts gradient flow, reducing learning efficiency and leading to unstable training dynamics. By relaxing this constraint and allowing global read operations, our differential stack achieves greater performance. A detailed discussion is provided in §B.4. Therefore, we propose the global read mechanism, by collecting information over the stack. Global read is achieved through a learnable query-over-stack attention:

$$\text{R}_t = \text{Softmax}(W_g \cdot (\text{St}_t \otimes \text{M}_t)) \cdot \text{St}_t \tag{4}$$

where $\otimes$ refers to element-wise production and $W_g \in \mathbb{R}^S$ is a trainable query vector. The Softmax term computes the attention score, and the content read from the stack is the weighted sum of all stack elements, where each element is weighted by its corresponding attention score. The final output is a residual-like connection $h_t' = g_h \cdot h_t + \text{R}_t$, where $g_h$ is a trainable parameter.

## 3.2 Multi-Head Low-Rank Stack

In the Transformer architecture, the multi-head attention mechanism [Vaswani et al., 2017] processes multiple attention patterns in parallel across different representation subspaces. The design enables

the model to capture diverse relationships within the input sequence. Following a similar design philosophy, we propose the multi-head low-rank stack. Specifically, we down-project the hidden state $h_t \in \mathbb{R}^d$ and split it into subspaces ($\mathbb{R}^{d_s}$) as: $\left[ h_t^{(1)}, h_t^{(2)}, \ldots, h_t^{(H)} \right] = \text{Reshape}(W_{\text{down}} \cdot h_t)$, where $H$ denotes the number of stack heads and $W_{\text{down}} \in \mathbb{R}^{(H \cdot d_s) \times d}$ is the down-projection matrix. *Reshape* is a standard operation in deep learning that changes the shape of a tensor. In this context, it reshapes the down-projected vector into $H$ independent vectors of dimension $d_s$ for the multi-head stacks. Each head corresponds to an independent stack as introduced in §3.1.

Given the down-projected hidden state $h_t^{(i)} \in \mathbb{R}^{d_s}$ for the $i$-th head, the stack element $\text{St}_t^{(i)}$ and the mask $\text{M}_t^{(i)}$ (both in $\mathbb{R}^{S \times d_s}$) along with the final read-out $\text{R}_t^{(i)}$ are computed independently for each head following Equations 1 - 4. After performing soft updates and global reads for all $H$ heads, we concatenate their outputs to obtain the final result: $h_t' = g_h \cdot h_t + W_{\text{up}} \cdot \text{Concat} \left( \text{R}_t^{(1)}; \cdots ; \text{R}_t^{(H)} \right)$, where $W_{\text{up}} \in \mathbb{R}^{d \times (H \cdot d_s)}$ is the up-projection matrix. This multi-head mechanism allows the model to organize stack operations into different patterns in parallel, thereby capturing diverse relationships and dependencies more effectively. Empirically, we find that a small $H$ (*e.g.*, 4 or 8) is sufficient to achieve notable performance. For more details on ablation studies and discussions, please refer to §6. On the other hand, the overall computational cost with the low-rank design is much smaller than counterpart of a single-head stack with full dimensions due to the reduced dimension of low-rank adaptation in each sub-stack. Refer to §6 for ablations.

### 3.3 Key Implementation & Training Know-Hows

The modular stack of STACKTRANS enables it to augment the Transformer architecture without altering the Transformer layers themselves. There are some key implementation and training insights.

**Stack Overflow** In §3.1 and §3.2, we assume that the stack size $S$ is sufficiently large or even infinite. However, due to limitations in computational power and storage resources, $S$ is typically relatively small, making overflow inevitable. In our implementation, we address this by truncating the stack and setting all overflow elements to zero, that is, $\text{St}_t[i] = \vec{0}$ if $i \geq S$. This truncation can be seen as a form of "forgetting", where the information carried by the overflow elements is discarded.

**Training parallelism** Ideally, the stack is supposed to process hidden states according to their temporal or layer dependencies, *i.e.*, it should prioritize hidden states generated from earlier tokens or shallower layers. One feasible sequence fed into the hidden state stack would be $[h_{t_0,0}, h_{t_0,1}, \cdots, h_{t_0,L}, h_{t_1,0}, \cdots]$, where $h_{t,i} \in \mathbb{R}^d$ denotes the hidden state of the $i$-th Transformer layer at token $t$, and $L$ represents the total number of layers. Although such a behavioral pattern is clearly beneficial for learning stack operations, the temporal dependencies conflict with the parallel training of the Transformer layers. To facilitate training parallelism, we implement STACKTRANS by breaking these temporal dependencies. Specifically, STACKTRANS learns stack operations based on the hidden state sequence at token $t_i$ from layer 0 to $L$ ($[h_{t_i,0}, \cdots, h_{t_i,L}]$), allowing all tokens to be trained in parallel. We provide detailed discussion in §B.3.

**Stack regularization** During training, STACKTRANS optimizes the standard autoregressive language modeling loss ($\mathcal{L}_{\text{LM}}$) over the token sequence. To prevent the operation probabilities $a_t$ from collapsing into uniform, we introduce an entropy-based regularization term, defined as $\mathcal{L}_{\text{St}} = \sum_t \mathcal{H}(a_t)$, where $\mathcal{H}(\cdot)$ calculates the entropy. The overall loss function combines the language modeling loss and the stack regularization term, $\mathcal{L} = \mathcal{L}_{\text{LM}} + \lambda \cdot \mathcal{L}_{\text{St}}$, where $\lambda$ is a hyperparameter. As a regularization term, we give $\lambda$ a small weight in experiments, e.g., 0.001.

## 4 Evaluation against Formal Languages

Understanding formal languages is fundamental for modeling many aspects of real-world natural language processing tasks. To highlight the motivation behind our stack-enhanced mechanism, we first evaluate STACKTRANS on formal language modeling tasks inspired by the Chomsky hierarchy.

**Experimental Setup** In this section, we evaluate STACKTRANS on three groups formal language modeling tasks aligned with the Chomsky hierarchy [Delétang et al., 2022]. These tasks assess a

Table 1: Test accuracy on formal language tasks compared to Transformer [Vaswani et al., 2017], LSTM [Graves and Graves, 2012], StackRNN [Joulin and Mikolov, 2015]) and StackAttn [Li et al., 2024].

| Task | LSTM | StackRNN | StackAttn[*] | Transformer | STACKTRANS |
|---|---|---|---|---|---|
| *Regular (RE) Tasks* | | | | | |
| Even Pairs | 1.00 | 1.00 | - | 0.49 | 1.00 |
| Parity Check | 1.00 | 1.00 | - | 0.50 | 1.00 |
| Cycle Navigation | 0.89 | 1.00 | - | 0.20 | 1.00 |
| *Deterministic Context-Free (DCF) Tasks* | | | | | |
| Stack Manipulation | 0.66 | 0.85 | 0.93 | 0.53 | 0.92 |
| Reverse String | 0.71 | 0.80 | 1.00 | 0.55 | 1.00 |
| Modular Arithmetic | 0.43 | 0.42 | 0.30 | 0.30 | 0.60 |
| *Context-Sensitive (CS) Tasks* | | | | | |
| Missing Duplicate | 0.68 | 0.67 | - | 0.53 | 1.00 |
| Odds First | 0.60 | 0.55 | - | 0.51 | 0.53 |
| Binary Addition | 0.56 | 0.51 | - | 0.48 | 0.48 |
| Binary Multiplication | 0.56 | 0.53 | - | 0.50 | 0.48 |
| Compute Sqrt | 0.64 | 0.63 | - | 0.51 | 0.57 |
| Bucket Sort | 0.79 | 0.75 | - | 0.79 | 0.89 |

[*] Results listed here are reported by Li et al. [2024].

model's ability to learn underlying compositional rules of formal languages and generalize to input lengths beyond those seen during training. Please refer to §D for details. Following prior work [Delétang et al., 2022], we implement STACKTRANS with relatively limited parameters. Concretely, we use five Transformer layers with $d = 64$. $H$ is set to 4 and $d_s$ is set to 8. We compare STACKTRANS to some representative baselines, including the standard Transformer [Vaswani et al., 2017], LSTM [Graves and Graves, 2012], StackRNN [Joulin and Mikolov, 2015]) and StackAttn [Li et al., 2024], maintaining identical experimental settings for all models.

**Evaluation Results** Table 1 shows that STACKTRANS consistently outperforms the standard Transformer, particularly on RE and DCF tasks. For RE tasks, most evaluated models attain near-perfect accuracy. However, Transformers tend to falter in the absence of explicit inductive biases. This further underscores the effectiveness of the hidden state mechanism introduced in STACKTRANS. When compared with the state-of-the-art stack-augmented approaches, such as StackRNN and StackAttn, STACKTRANS either outperforms them or is at least on par with them in nearly all tasks.

The hidden state stack mechanism likely endows STACKTRANS with characteristics akin to pushdown automata. This is evidenced by its superior performance over all baselines on both RE and DCF tasks, which are known to be solvable by pushdown automata. However, as pushdown automata are theoretically incapable of resolving CS tasks, we observe that all approaches, including STACKTRANS, perform poorly on CS tasks. Despite this limitation, STACKTRANS demonstrates the ability to handle a subset of CS tasks, which we attribute to specific design enhancements such as the multi-head mechanism and the global reading capability. These features provide STACKTRANS with stronger modeling capacity than traditional pushdown automata, enabling it to capture additional dependencies and complexities beyond the theoretical limits of pushdown automata.

## 5 Evaluation against General Natural Languages

From the perspective of computational linguistics, natural languages are generally considered to belong to a class of languages that includes DCF [Gazdar and Pullum, 1982; Chomsky, 1956; Shieber, 1985]. Therefore, to thoroughly assess STACKTRANS, we conduct further evaluations on general language modeling tasks. ❶ We examine the scalability of STACKTRANS through scaling law studies, analyzing the effect caused by model size and training tokens. ❷ A STACKTRANS with 360 million parameters is pretrained with nearly 980 billion tokens. We evaluate its performance on a variety of standard benchmarks, comparing it to models with similar or larger parameter scales. ❸ We provide additional empirical observations through ablation studies and deeper analysis (please refer to §6).

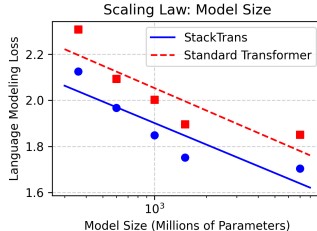
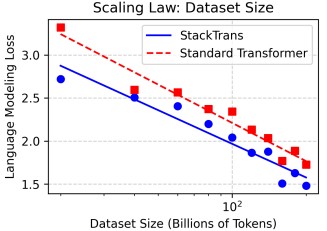
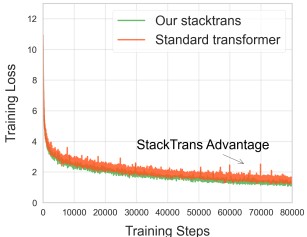

| (a) Scaling law on model size | (b) Scaling law on dataset size | (c) Training loss dynamics |
| --- | --- | --- |

Figure 3: Scaling law and training loss dynamics across early steps of STACKTRANS.

Table 2: Evaluation results on natural language tasks.

|  | STACKTRANS | SmolLM | SmolLM2 | Qwen2.5 | OLMo | TinyLLaMA |
| --- | --- | --- | --- | --- | --- | --- |
| Param. # (B) | 0.36 | 0.36 | 0.36 | 0.5 | 1.1 | 1.1 |
| Token # (T) | 1 | 0.6 | 4 | 18 | 2 | 3 |
| HellaSwag | 59.5 | 51.8 | 54.5 | 51.2 | 60.7 | 55.2 |
| ARC | 59.0 | 50.1 | 53.0 | 45.4 | 44.0 | 43.4 |
| PIQA | 71.7 | 71.6 | 71.7 | 69.9 | 75.2 | 72.9 |
| MMLU | 36.5 | 34.4 | 35.8 | 33.7 | 31.9 | 32.2 |
| CommonsenseQA | 37.1 | 35.3 | 38.0 | 31.6 | 40.3 | 37.0 |
| TriviaQA | 11.2 | 9.1 | 16.9 | 4.3 | 2.8 | 9.8 |
| Winogrande | 52.8 | 52.8 | 52.5 | 54.1 | 53.2 | 55.7 |
| OpenBookQA | 37.5 | 37.2 | 37.4 | 37.4 | 38.0 | 33.2 |
| GSM8K (5-shot) | 33.6 | 1.6 | 3.2 | 33.4 | 1.8 | 1.7 |
| Average | **44.3** | 38.2 | 40.3 | 40.1 | 38.7 | 37.9 |

**Experimental Setup** We follow the OLMo framework [AllenAI, 2024] to pretrain STACKTRANS. Our corpora come from Dolma [Soldaini et al., 2024] and Smoll [Allal et al., 2025], which contain high-quality natural language, math, and Python code examples with diverse domains. We carry out data filtration, ultimately obtaining approximately 980 billion tokens. The data filtering followed standard LLM pre-training procedures, including deduplication, removal of low-quality text, and filtering of harmful content. To scale up model parameters, we adapt the Dolma v1.6-sample configuration in OLMo, using roughly 80 billion tokens for each variant model training. For scaling up training tokens, we train STACKTRANS with 360M parameters on a sampled subset of 200 billion tokens from pretraining corpora.

**Scaling Law of STACKTRANS** We train STACKTRANS models with a range of parameter sizes (360M, 600M, 1.0B, 1.5B, and 7B) under the same training budget in terms of tokens. The language modeling loss is tracked throughout the training process, and the scaling trends are depicted in Figure 3. Our observations find that STACKTRANS exhibits smoother convergence and attains lower final loss compared to standard Transformers of equivalent size. Notably, even with 360M parameters, STACKTRANS consistently demonstrates smaller loss, which underscores the significant contribution of the hidden state stack mechanism to improved generalization capabilities. Overall, STACKTRANS aligns well with the predicted scaling trends and delivers superior performance. To analyze the optimization process, we compare training loss dynamics between STACKTRANS and the standard Transformer in Figure 3(c). Detailed analysis is shown in §B.5.

**Evaluation against STACKTRANS-360M** We pre-train STACKTRANS-360M from scratch, and the detailed model configuration is shown in §F. To assess the downstream capabilities, we evaluate STACKTRANS-360M on a comprehensive suite of widely-used benchmarks, and details are shown in §E. As listed in Table 2, STACKTRANS-360M outperforms all baseline models, including those with significantly larger parameter sizes. Notably, it achieves substantial gains on GSM8K and ARC, highlighting its strength in reasoning tasks that require compositional generalization, recursion, or latent state management. Despite having fewer parameters, STACKTRANS performs competitively on PIQA and CommonsenseQA, further indicating that the stack-augmented memory module improves

Table 3: Training and validation results of stack ablations (~20B tokens).

| Model | Train Loss ↓ | V2 Loss ↓ | V2 PPL ↓ | V3 Loss ↓ | V3 PPL ↓ |
|---|---|---|---|---|---|
| Transformer | 2.411 | 3.518 | 34.38 | 3.195 | 25.33 |
| STACKTRANS | 2.359 | 3.432 | 32.89 | 3.092 | 24.50 |
| QueueTrans (Stack→Queue) | 2.679 | 3.679 | 35.14 | 3.211 | 25.97 |
| Push-Only (Fix $a_t^{push} = 1$) | 2.875 | 4.032 | 39.02 | 3.407 | 27.13 |
| Single-Head ($H = 1$) | 2.493 | 3.552 | 33.56 | 3.130 | 24.91 |
| Full-Dimension ($H \cdot d_s = d$) | 2.370 | 3.457 | 33.05 | 3.105 | 24.73 |

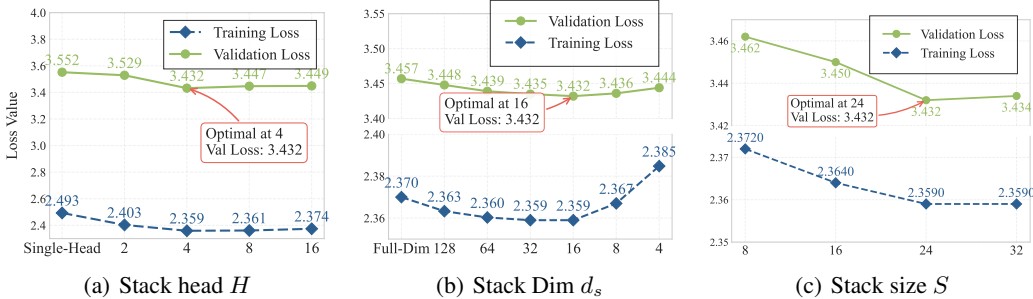

(a) Stack head $H$      (b) Stack Dim $d_s$      (c) Stack size $S$

Figure 4: Hyperparameters ablations of STACKTRANS.

representation capacity without compromising generalization. Overall, STACKTRANS-360M achieves an average performance of 44.3 across 9 diverse tasks, exceeding comparable models in the table with only a fraction of the parameter size and dataset size. Our proposed STACKTRANS enhances LLM's generalization ability, especially in scenarios with limited computation and parameter budgets.

## 6 Discussion

**Ablations of Key Designs** To assess the impact of the stack design in STACKTRANS, we investigate two alternative configurations. ❶ We replace the stack with a queue, adopting a first-in-first-out storage structure, which we term QueueTrans. Apart from this modification, the queue mask performs similar functions to the stack mask in maintaining valid and activated elements. ❷ In this extreme setting, we modify the stack in STACKTRANS to "push-only", where $a_t^{push}$ is fixed to 1. This configuration essentially disables pop and no-op operations. The multi-head stack and the low-rank compression are the other two crucial design elements in STACKTRANS. To study their impact, we conduct ablation studies as follows: ❸ We disable the multi-head splitting, reverting it back to a single-head stack. ❹ We remove the down- and up-projections (low-rank mechanism) from the full stack. In total, we create four variants for ablation studies.

We evaluate all the variants introduced above on the V2 and V3 validation sets [Zhu et al., 2024]. Experimental details are provided in §G. The ablation results are presented in Table 3. The QueueTrans variant exhibits notably higher perplexity and lower overall accuracy compared to STACKTRANS, particularly on tasks involving hierarchical or recursive patterns. This outcome is consistent with our expectation that queue operations are inherently less effective at modeling nested dependencies and grammars. Similarly, the push-only variant performs poorly, with both training and validation losses significantly deteriorating. The absence of pop operations impairs its ability to dynamically manage and retrieve stored information, thereby reducing its overall effectiveness. Both single-head and full-dimensional variants are consistently outperformed by STACKTRANS. The multi-head mechanism enhances flexibility by enabling parallel decomposition of stack streams, while the low-rank mechanism reduces computational costs without compromising much modeling capacity. The results show that the single-head stack is more efficient but performs slightly worse, whereas the full-dimension stack is extremely costly for limited performance gain. This precisely demonstrates that our multi-head, low-rank design is the optimal trade-off, achieving most of the performance gains at a cost close to the baseline Transformer.

**Ablations of Hyperparametesr**    In addition to the configuration of the standard Transformer layers, STACKTRANS has three key hyperparameters – the number of stack heads $H$, the dimension of each stack head $d_s$, and the stack size $S$. We perform grid search over reasonable ranges for these hyperparameters, training STACKTRANS with 20 billion tokens. The curves of training loss and validation loss (evaluated on V2 [Zhu et al., 2024]) are plotted in Figure 4. It is clear that $H$ is crucial for parallelism, but Figure 4(a) indicates that performance plateaus once $H$ surpasses a certain threshold. From Figure 4(b), we observe that setting $d_s$ within the range from 16 to 64 balances the computational cost and the model's expressiveness effectively. Similarly, Figure 4(c) shows that increasing $S$ from 24 to 32 has nearly no impact on performance. Given that a larger $S$ leads to higher storage overhead, we make a trade-off and ultimately set $S$ to 24 during our evaluation.

**More Detailed Discussion**    Due to the length constraints of the paper, we provide more discussions in the appendices. For in-depth investigations of STACKTRANS's training and inference efficiency, please refer to §B.1. In our implementation, we break the temporal dependencies to facilitate training parallelism of STACKTRANS, as briefly introduced in §3.3. We further discuss why this approximation works in §B.3. We adopt global reading rather than top peeking for STACKTRANS, and we explain the rationale behind this design in §B.4. We provide visualizations of the stack action patterns across different tasks in §B.2, and analyze the training dynamics in §B.5.

# 7    Conclusion

Inspired by pushdown automata, we propose STACKTRANS, a novel Transformer variant architecture integrating differentiable hidden state stacks in between Transformer layers. STACKTRANS improves generalization in both formal language tasks and natural language modeling tasks. In particular, our from-scratch pretrained STACKTRANS-360M outperforms several larger open-source LLMs with 2–3× more parameters, showcasing its superior efficiency and reasoning capability.

# Acknowledgments and Disclosure of Funding

This research is supported by the National Key R&D Program under Grant No. 2023YFB4503801, the National Natural Science Foundation of China under Grant No. 62192731, 62192730, 62192733, the Major Program (JD) of Hubei Province (No.2023BAA024).

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

# A    Related Work

**Evolvement of Large Language Models**    The success of LLMs is deeply rooted in the Transformer architecture [Vaswani et al., 2017] and its subsequent variations, which serve as the backbone for LLMs such as GPT [Radford et al., 2018; GPT-4, 2023; GPT-4o, 2024], DeepSeek [Bi et al., 2024; Liu et al., 2024], LLaMA [Touvron et al., 2023], and other prominent LLM series [Brown et al., 2020]. Scaling laws [Kaplan et al., 2020] have shown that increasing model parameter size leads to emergent capabilities, enabling LLMs to tackle increasingly complex tasks and exhibit surprising generalization behaviors. In addition to proprietary closed systems, open-source initiatives like OLMo [Groeneveld et al., 2024] demonstrate the potential of community-scale pretraining using meticulously curated datasets such as Dolma [Soldaini et al., 2024]. These efforts highlight how transparent methodologies can accelerate innovation, making powerful LLMs more accessible for academic research and industrial applications. Despite their widespread adoption, the Transformer architecture has inherent expressivity limitations [Hahn, 2020]. Although Transformers are theoretically Turing complete [Chomsky, 1956], they often underperform on tasks tied to formal languages within the Chomsky hierarchy. Recent studies [Delétang et al., 2022; DuSell and Chiang, 2024] have shown that standard Transformers struggle with recursive and hierarchical patterns across both synthetic and real-world tasks. These limitations underscore the need for fundamental architectural enhancements to better model the Chomsky hierarchy, such as rich linguistic and algorithmic structures.

Building on prior work and drawing inspiration from pushdown automata [DuSell and Chiang, 2024; Li et al., 2024; Sartran et al., 2022], we address these limitations by introducing hidden state stacks into the Transformer architecture. Our proposed method, STACKTRANS, enhances the model capacity to represent hierarchical dependencies and recursive grammars, enabling it to learn Chomsky hierarchy grammars effectively. We believe that fostering transparent and open discussions around the underlying architectural challenges will accelerate the evolution of Transformer-based models and propel the development of large language models to new heights.

**Stack Augmentation**    Equipping neural networks with external data structures, such as stacks, has been widely explored to enhance models' ability to recognize hierarchical and context-free languages. Although earlier studies primarily focused on recurrent neural networks [Joulin and Mikolov, 2015; Grefenstette et al., 2015], recent efforts have adapted these thoughts to Transformer-based models [DuSell and Chiang, 2024; Li et al., 2024; Sartran et al., 2022]. They aim to embed stack-like operations into Transformers to address their shortcomings in modeling the Chomsky hierarchy, particularly DCFs. For example, Li et al. [2024] augment standard attention layers with differentiable stacks, enabling soft push/pop operations to model recursive structures. However, this comes with architectural trade-offs, where the stack control is tightly coupled with the attention mechanism, leading to increased entanglement and reduced modularity. Mali et al. [2019] and Stogin et al. [2024] develop a rigorous theory: they prove orbital stability of continuous-stack encodings and show that finite-precision RNNs equipped with such stacks are Turing-complete and can simulate any PDA/TM—providing tight neuron bounds. Further advances, such as those in DuSell and Chiang [2024], embed stack operations directly within attention heads, providing stronger inductive biases. While these approaches better model Chomsky hierarchy grammars, their validation is largely limited to small-scale models and synthetic datasets. It raises questions about their scalability and

Table 4: Training and inference efficiency, including time cost and GPU memory usage.

| Model Variant | Time Cost | | Peak GPU Memory Usage |
|---|---|---|---|
| | Training | Inference | |
| Transformer[*] | ×1.00 | ×1.00 | ×1.00 |
| STACKTRANS | ×1.16 | ×1.09 | ×1.12 |
| Single-Head Stack | ×1.03 | ×1.04 | ×1.00 |
| Full-Dimensional Stack | ×3.78 | ×3.30 | ×1.73 |

[*] All values represent multiples of results of the baseline Transformer.

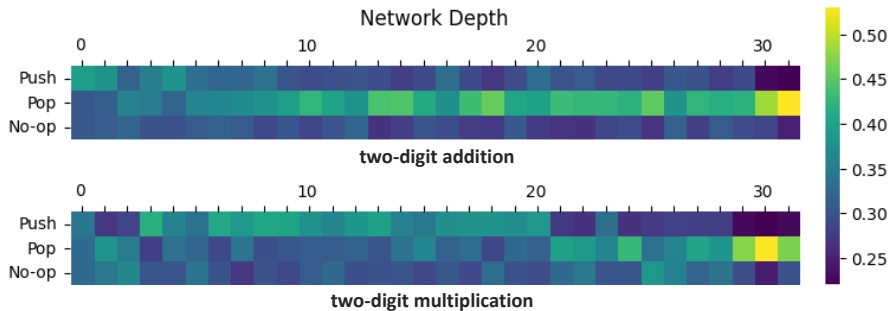

Figure 5: Average probabilities of three operations across the network depth for the two-digit addition task and the two-digit multiplication task.

generalization ability to large-scale tasks and larger Transformer-based models with millions or billions of parameters. In contrast, STACKTRANS introduces a differentiable stack. Rather than tempering the attention mechanism, STACKTRANS integrates stacks in between standard Transformer layers. This design decouples stack manipulation from attention computations, enabling seamless integration with pretrained Transformer models while preserving compatibility across architectures of varying sizes (ranging from 0.36B to 7B parameters). By focusing on hidden states, STACKTRANS maintains flexibility and scalability, addressing broader linguistic and algorithmic challenges without compromising the core principles of the Transformer architecture.

# B  Discussion (Cont.)

Following §6, we provide some more discussions in this section.

## B.1  Training and Inference Efficiency

Considering that the training and inference efficiency is an important factor for model applications. In this section, we investigate the training and inference efficiency of STACKTRANS compared to the standard Transformer. To keep a fair comparison, all comparisons are conducted on the same hardware setup. We measure both training and inference time over 100 consecutive steps under identical hyperparameter and batch size configurations. To explore the design trade-offs, we also compare several STACKTRANS variants, including single-head stack and full-dimensional stack as described in §6. As shown in Table 4, despite incorporating differentiable stack modules, STACKTRANS achieves competitive training and inference efficiency. Concretely, it introduces only marginal overhead (around 10%) compared to the standard Transformer while yielding significant performance improvements. Besides, the memory usage increase is moderate, well within the typical consumption range of large-scale LLM. This suggests that STACKTRANS offers a practical and scalable approach, and its stack mechanism can be integrated into existing Transformer without compromising deployment efficiency.

## B.2 Stack Action Patterns across Layers and Tasks

STACKTRANS introduces layer-wise stack modules that manipulate memory using three soft actions: push, pop, and no-op. Since stack dynamics play a central role in the model's expressiveness, we conduct an in-depth analysis of the stack action patterns across layers and downstream tasks. We select two arithmetic tasks from our synthetic benchmark suite: the two-digit addition task and the two-digit multiplication task. Although both tasks involve structured numerical reasoning, multiplication generally requires deeper or more nested intermediate steps than addition. Our STACKTRANS-360M achieves 100% accuracy on both tasks, indicating the performance of our model on these basic arithmetic questions. For every layer and timestep, we compute the average action probabilities of the three operations for each Transformer layer in STACKTRANS and visualize their trends across the network depth, as shown in Figure 5.

The results reveal a consistent action distribution pattern on both tasks: earlier layers predominantly favor push operations, while later layers exhibit an increased use of pop, with no-op remaining relatively stable throughout. This trend suggests that STACKTRANS automatically learns to incrementally store information during lower layers and retrieve it in upper layers. The intuitive behavior mirrors how hierarchical or recursive structures are processed.

Figure 5 further shows that multiplication elicits markedly more push operations in middle layers and deferred pop activity in higher layers, reflecting the deeper computation graph required by the task. In contrast, addition induces a flatter push/pop pattern distributed more evenly across layers, consistent with its shallower reasoning structure. These findings confirm that STACKTRANS learns to adapt memory access patterns dynamically according to task complexity, and that its stack behavior is both interpretable and task-sensitive.

## B.3 Approximations for Training Parallelism

In our implementation, we introduce necessary approximations to maintain training parallelism while preserving model performance, as detailed in §3.3. Temporal dependencies among hidden states are a fundamental aspect of the Transformer's processing pipeline. Let $\mathbf{h}_{t,i} \in \mathbb{R}^d$ represent the hidden state at layer $i$ for token $t$, where the ideal processing order for our stack would follow the complete sequence:

$$[\mathbf{h}_{t_0,0}, \mathbf{h}_{t_0,1}, \ldots, \mathbf{h}_{t_0,L}, \mathbf{h}_{t_1,0}, \ldots], \tag{5}$$

with $L$ denoting the total number of layers. However, to enable practical training parallelism, we introduce a controlled truncation between $\mathbf{h}_{t,L}$ and $\mathbf{h}_{t+1,0}$. This approximation allows us to compute token losses for all elements in a sequence simultaneously, which would otherwise be computationally prohibitive.

The decision to break these temporal dependencies is guided by two key considerations. First, the self-attention mechanism inherently captures cross-token relationships, which can partially compensate for the truncation. Second, the stack mechanism introduced in our model complements the attention layers by retaining sequential dependencies through external memory operations. Empirical evidence from prior work [Kudo et al., 2024] supports this design, showing that those intermediate-layer hidden states for subsequent tokens effectively preserve information from earlier tokens. Overall, this approximation achieves an optimal trade-off between computational efficiency and model performance, enabling scalable and parallelizable training while maintaining our designed stack mechanism.

## B.4 Global Reading Capability

Traditional stacks only permit access to the top element. However, in neural network modeling, such a restriction is unnecessary since tensor vectors can be efficiently accessed through operations like matrix multiplication. To enhance the stack's representational ability, our differential stack eliminates this constraint by introducing global reading capabilities and enabling full random access.

Furthermore, we find that enforcing a strict top-only access during training leads to unstable and suboptimal model performance. We attribute this to the frequent stack operations in neural networks: limiting access to the top element disrupts gradient flow and reduces parameter learning efficiency. By relaxing this constraint and enabling global read operations, our differential stack achieves greater representational ability and improved adaptability across diverse tasks.

### B.5  Training Loss Dynamics

To analyze the optimization process, we conduct a comparative analysis of training loss dynamics between STACKTRANS and the standard Transformer. The training curves are presented in Figure 3(c). One may find out that STACKTRANS exhibits a slightly slower decrease in loss during the very early training stages compared to the standard Transformer. We attribute this behavior to the additional learning complexity introduced by the hidden state stack, where STACKTRANS must learn when and how to carry out stack operations. As training progresses, however, STACKTRANS not only catches up but eventually surpasses the standard Transformer in convergence speed. Once the stack operation distribution stabilizes, STACKTRANS begins to leverage the stack more effectively, leading to a steeper decline in loss and an overall lower convergence plateau. This phenomenon shows that STACKTRANS, while requiring a slightly longer warm-up phase, ultimately achieves greater learning efficiency and a superior asymptotic performance ceiling compared to the standard Transformer.

## C  Limitations

While STACKTRANS demonstrates strong performance across a variety of tasks, there are several limitations to our work that we aim to address.

**Limitations of Model Size and Dataset Size**   Constrained by computational resources, we limit our final pre-trained model to 360M parameters and use approximately 1 trillion training tokens. Although the results show competitive performance, the scaling law discussed in §5 suggests that larger models and datasets could further amplify the strengths of STACKTRANS. Particularly, scaling up the number of model parameters and training tokens may enhance its ability to tackle more complex tasks. This limitation highlights the importance of access to large-scale computing infrastructure for future research. We hope to leverage the power of the open-source community to validate this new architecture.

**Necessary Approximation in Design**   To achieve training parallelism, we introduce controlled approximations in the sequence processing pipeline, as detailed in §3.3. Specifically, the truncation of temporal dependencies between tokens facilitates scalable training but may reduce the model's ability to fully exploit fine-grained sequential patterns. While the self-attention mechanism and stack-based memory mitigate this limitation, the truncation approximation may still pose risks, especially in tasks requiring deep inter-token dependencies. We plan to explore more robust and efficient settings in future work.

## D  Formal Language Task Details

We follow the experimental settings of Delétang et al. [2022]. Table 5 presents an overview of the formal language tasks and their complexity level within the Chomsky hierarchy. These tasks assess models' ability to learn underlying compositional rules of formal languages and generalize to input lengths beyond those seen during training. The three task groups are categorized by the Chomsky hierarchy type, including RE (type-3), DCF (type-2), and CS (type-1). The classification adheres to formal automata theory, associating the three tasks with finite-state automata, pushdown automata, and linear-bounded automata, respectively. Please refer to Table 5 in §D for definitions and examples of these tasks. Despite the presence of classification tasks, all tasks are formulated as sequence mapping problems. In this setup, the model takes an input sequence and decodes it into an output sequence. STACKTRANS is trained on sequences with input length uniformly sampled from 1 to 40 tokens. At test time, we evaluate STACKTRANS on sequences with significantly longer lengths up to 500 tokens, thereby measuring its length generalization. Following the same procedure as Delétang et al. [2022], token-level accuracy is used as the evaluation metric. We repeat each experimental configuration ten times and report the best accuracy achieved.

## E  General Natural Language Task Details

To assess the downstream capabilities, we evaluate STACKTRANS-360M on a comprehensive suite of widely-used benchmarks [Brown et al., 2020; Touvron et al., 2023; Groeneveld et al., 2024], including those for common sense reasoning (*e.g.*, HellaSwag, PIQA) [Zellers et al., 2019; Clark

Table 5: Formal language task descriptions and input-output examples.

| Task Name | Description |
|---|---|
| *Regular (RE) Tasks* | |
| Even Pairs | Check if the count of `ab`/`ba` pairs is even. |
| Parity Check | Check if the count of `b` is even. |
| Cycle Navigation | Navigate movements on a modulo-5 cycle. |
| *Deterministic Context-Free (DCF) Tasks* | |
| Stack Manipulation | Perform stack operations and return the final state. |
| Reverse String | Reverse the input string using a stack. |
| Modular Arithmetic | Evaluate nested arithmetic expressions modulo 5. |
| Solve Equation | Find a variable satisfying a modular equation. |
| *Context-Sensitive (CS) Tasks* | |
| Binary Addition | Compute binary addition of two numbers. |
| Binary Multiplication | Compute binary multiplication. |
| Compute Sqrt | Compute the integer square root of a binary number. |
| Bucket Sort | Sort a sequence over a fixed alphabet. |
| Duplicate String | Output the string concatenated with itself. |
| Missing Duplicate | Find the missing character in a duplicated string. |
| Odds First | Interleave odd and even indices of a sequence. |

| Task Name | Input Example | Output Example |
|---|---|---|
| *Regular (RE) Tasks* | | |
| Even Pairs | `aabba` | True |
| Parity Check | `aaabba` | True |
| Cycle Navigation | `011210` | 2 |
| *Deterministic Context-Free (DCF) Tasks* | | |
| Stack Manipulation | `abbaa POP PUSH a POP` | `abba` |
| Reverse String | `aabba` | `abbaa` |
| Modular Arithmetic | $-(1-2) \cdot (4-3 \cdot (-2))$ | 0 |
| Solve Equation | $-(z-2) \cdot (4-3 \cdot (-2)) = 0$ | 1 |
| *Context-Sensitive (CS) Tasks* | | |
| Binary Addition | `10010 + 101` | `10111` |
| Binary Multiplication | `10010 × 101` | `1001000` |
| Compute Sqrt | `101001` | `101` |
| Bucket Sort | `421302214` | `011222344` |
| Duplicate String | `abaab` | `abaababaab` |
| Missing Duplicate | `ab_aba` | `a` |
| Odds First | `aaabaa` | `aaaaba` |

et al., 2018; Hendrycks et al., 2021; Bisk et al., 2020; Sakaguchi et al., 2020], question answering (*e.g.*, OpenBookQA) [Talmor et al., 2019; Joshi et al., 2017; Mihaylov et al., 2018], and math-based reasoning (GSM8K) [Cobbe et al., 2021] [2]. We compare STACKTRANS-360M with other open-source models around 1B parameters, including SmolLM-360M [Allal et al., 2024], SmolLM2-360M [Allal et al., 2025], Qwen2.5-0.5B [Yang et al., 2024], OLMo-1B [Groeneveld et al., 2024], and TinyLLaMA-1B [Zhang et al., 2024b]. We use the **lighteval** framework [huggingface, 2024], and for all applicable tasks, we adhere to zero-shot evaluation settings, unless otherwise specified.

---

[2]The evaluation protocol strictly follows Allal et al. [2025], and we obtain nearly identical results to those reported in the paper.

Table 6: Model configuration of STACKTRANS-360M.

| Parameter | Value |
|---|---|
| Vocabulary Size | 49152 |
| Number of Attention Heads | 15 |
| Number of Hidden Layers | 32 |
| Hidden Size | 960 |
| Intermediate Size (FFN) | 2560 |
| Attention Dropout | 0.0 |
| Activation Function | Silu |
| Number of Stack Heads | 4 |
| Stack Dimensionality | 16 |
| Stack Size | 24 |
| Maximum Position Embeddings | 4096 |
| RoPE Scaling | None |
| RoPE $\theta$ | 100000 |

Table 7: Overview of V2 and V3 Validation Sets. Each validation set includes diverse text sources to ensure comprehensive evaluation.

| Validation Set | Datasets Included |
|---|---|
| **V2 Validation Sets** | v2-small-4chan-validation, v2-small-c4_100_domains-validation, v2-small-c4_en-validation, v2-small-gab-validation, v2-small-ice-validation, v2-small-m2d2_s2orc-validation, v2-small-m2d2_wiki-validation, v2-small-manosphere-validation, v2-small-mc4_en-validation, v2-small-pile-validation, v2-small-ptb-validation, v2-small-twitterAEE-validation, v2-small-wikitext_103-validation |
| **V3 Validation Sets** | v3-small-c4_en-validation, v3-small-dolma_books-validation, v3-small-dolma_common_crawl-validation, v3-small-dolma_pes2o-validation, v3-small-dolma_reddit-validation, v3-small-dolma_stack-validation, v3-small-dolma_wiki-validation, v3-small-ice-validation, v3-small-m2d2_s2orc-validation, v3-small-pile-validation, v3-small-wikitext_103-validation |

# F   Model Configuration of STACKTRANS-360M

Table 6 shows the detailed model configuration of our STACKTRANS-360M, which is inspired by the similar setting in Allal et al. [2024] and Allal et al. [2025]. The stack-related setting is decided by a grid search in §6.

# G   Validation Dataset Details for General Language Modeling

Following the method of Zhu et al. [2024], we evaluate all variants on the **V2 Validation Sets** and **V3 Validation Sets** curated within the OLMO framework. The specific datasets for V2 and V3 validation [Zhu et al., 2024] are shown in Table 7.

