# OpenReview forum: "Recursive Transformer: Boosting Reasoning Ability with State Stack"
_NeurIPS.cc/2025/Conference — NeurIPS 2025 poster_

### Official Review · Reviewer_j9mp · 2025-07-01

**Clarity:** 4
**Significance:** 3
**Originality:** 3
**Rating:** 5
**Confidence:** 4

**Summary:**

This paper proposes to solve the limitations of the standard transformers when processing formal languages of the Chomsky hierarchy.
Due to the equivalence of the pushdown automata (PDA) and the context-free grammar (CFG), the authors design a differentable hidden stack between consecutive transformer layers, yielding the StackTrans model. The setting injects the inductive bias of the way the PDAs work, and enhances the ability of processing grammars of the Chomsky hierarchy.

**Questions:**

According to the strengths and weaknesses above, some concerns are below:

- Can the learnt stack be shown having effectively captured the grammar?
- Is there any result of the StackTrans model in different sizes, and are they significant?
- Is there any performance degeneration of StackTrans on general natural language datasets where grammars of the Chomsky hierarchy are not that important?

**Ethical Concerns:**

["NO or VERY MINOR ethics concerns only"]

**Final Justification:**

My concerns listed in the weaknesses have been carefully addressed by the authors. I believe the overall quality is sufficient to make a score raising.

**Limitations:**

yes

**Quality:**

3

**Strengths And Weaknesses:**

Strengths.

- The proposed hidden stack module is more elegant than previous works and can work with other transformer designs with much less invasion in the existing codes. For example, the stacks can be neatly used with Flash Attentions.
- The StackTrans model is also shown satisfying the scaling law on model sizes and dataset sizes.
- Experiments have shown the effectiveness of StackTrans on several formal language tasks and some Natural Language Tasks.

Weaknesses.

- To some extent it remains unknown if the hidden stack is indeed a stack. For example, can we restore and check the grammar bottom-up, from the trained dataset? But in StackTrans, the stack is globally read, more like a global memory instead of a stack.
- The model details are not given. Since there are lots of variants of a transformer layer, could StackTrans work with most of them? If so, perhaps adding the hidden stack into LLMs of different families are needed. Furthermore, can we use the proposed hidden stack as the way of using LoRA?
- The results of different sizes of StackTrans are not given.
- Though the CFG grammar had been considered as one way to develop natural language grammars, it's also been widely admitted that there're lots of long-context dependencies in natural languages which cannot be described in CFG. Despite the long debates, at least we hope to see a CFG/PDA-inspired inductive bias for the neural models will not hurt the general performance on common corpus. So perhaps the StackTrans model is also required to be evaluated on general tasks (for example, the GLUE benchmark)

---

> ### Author Rebuttal · Authors · 2025-07-31
>
> We would like to sincerely thank the reviewer j9mp for their thorough and highly constructive feedback. We will address questions here in detail one by one.
> - On whether the "hidden stack is really a stack" and if "the learned grammar can be inspected."
> This is a very central question, and we will respond from two perspectives:
>   - On the name and the global read mechanism: We call it a "stack" based on its core update mechanism. The information flow is dynamically managed through differentiable push and pop operations. The "global read" you observed is a reading mechanism we introduced to solve a key problem in practical training. As described in Appendix §B.4, a traditional "stack-top only" read can lead to restricted gradient flow and unstable training. Therefore, our design can be seen as: a module with stack-like update logic, supplemented by a more flexible global read mechanism to optimize the learning process.
>   - On whether we can show that it captures grammar: Yes, we can.
>     - Quantitative Evidence: In the formal language tasks in §4, StackTrans achieved good performance on tasks requiring precise grammar. For example, it scored 1.00 on the "Reverse String" task, whereas the standard Transformer only scored 0.55. This directly demonstrates that it has learned the underlying grammatical rules.
>     - Qualitative Evidence: In Appendix §B.2 (Figure 5), we visualized the stack's operational patterns. The results clearly show that the model automatically learns meaningful, task-aligned patterns: it tends to push (store information) in the shallower layers of the network and pop (retrieve information) in the deeper layers. This interpretable behavior could be seen as the evidence that it has learned the underlying "grammar" of the task.
> - On model details, compatibility with variants, and comparison with LoRA:
> One of the core advantages of our design is its modularity and non-intrusive nature.
> As stated in the paper, StackTrans "integrates a differentiable stack between Transformer layers while maintaining the integrity of the Transformer layers" and "decouples the stack operations from the attention computation."This design allows it to be naturally combined with various Transformer backbones. Therefore, we have strong reason to believe that it can also be combined with parameter-efficient fine-tuning (PEFT) methods like LoRA (e.g., by freezing the backbone LLM and fine-tuning only the StackTrans modules). This is a very promising direction for future research.
> - On providing results for different model sizes and their significance:
> The reviewer may have overlooked the relevant sections, as we have provided detailed results in the paper:
>   - Scaling Law Study: In §5, we systematically studied the model's scaling laws, clearly analyzing the impact of both model size and the amount of training data.
>   - Performance Curves for Different Sizes: Figure 3(a) visually presents the loss curves as the model size increases from 360M to 7B, clearly showing that larger model sizes lead to better performance and outperform the standard Transformer.
>   - Evaluation of a Specific Model Size: All comparisons in Table 2 are centered around our StackTrans-360M model, which is compared against models with larger parameter counts (such as the 1.1B OLMo and TinyLLaMA). The results are significant.
> - On CFG bias and performance on general-purpose natural language tasks:
> Your question—"whether introducing a CFG bias would harm general-purpose performance"—is a core question that all methods with structured inductive biases must answer. We address this challenge head-on by evaluating our model on a wide range of general-purpose tasks.
>   - It did not harm performance; it improved it. As shown in Table 2, we tested on 9 diverse general-purpose benchmarks, including commonsense reasoning (e.g. HellaSwag), question answering (e.g. OpenBookQA), and mathematical reasoning (e.g. GSM8K).
>   - The results are significant. The outcomes show that not only did StackTrans not suffer from performance degradation, but it actually achieved significant performance gains. Our 360M model's average score (44.3) far surpasses baseline models that are 2-3 times its size in parameter count. This proves that our stack mechanism acts as an effective supplementary module that enhances the model's structured reasoning capabilities without sacrificing its general language abilities. While GLUE is a good benchmark, we believe the existing 9 benchmarks (for example, MMLU is a popular benchmark that is adopted by many pretraining LLM studies) already provide strong evidence of the model's generality.

---

> > ### Comment · Reviewer_j9mp · 2025-08-04
> > **Responses to the Author Rebuttal**
> >
> > Thanks for the very detailed and accurate explanation to the questions. I believe the concerns have been carefully resolved and I will raise my ratings.
> > I am curious on the mdoel configuration, which is related to the 3rd weakness above.
> > Are the Stack heads $H$ and size $d_s$  restricted to follow that $H * d_s = d$? However, since the StackTrans-360M has the hidden size $d=64$ (Sec. 4), and the $d_s$ can be 128 (Fig. 4(b)), the stack size is not bounded by the Transformer hidden size. Is this understanding correct?

---

> ### Author Response · Authors · 2025-08-06
> **Thanks for your comment.**
>
> Thank you for the positive feedback and for your decision to raise your rating. We are delighted that our responses have resolved your concerns and appreciate the opportunity to clarify points on the model configuration.
>
>
> You are correct -- the total stack dimensionality (H * d_s) is not required to be equal to the Transformer's hidden size (d). This is a deliberate and central part of our model's architecture.
>
> This is due to our low-rank design. Our standard StackTrans model intentionally uses a bottleneck where the total stack dimension is smaller than the model's hidden size (i.e., $H \cdot d_s < d$). This is achieved by first down-projecting the input hidden state before it is processed by the parallel stacks. The configuration where $H \cdot d_s = d$ is a special case we refer to as the "Full-Dimension" variant, which we tested in our ablation studies (Table 3). Those results confirmed that the full-dimension approach was significantly more computationally expensive without a corresponding performance benefit, validating our choice of the more efficient low-rank design.
>
> To further prevent any misunderstanding, we wish to highlight the two distinct experimental configurations used in our paper:
> - Section 4 (Formal Language Evaluation): For these more theoretical tasks, we used a smaller model configuration to ensure a fair comparison with prior work. This model uses five Transformer layers with a hidden size of d=64, 4 stack heads, and a stack dimension of d_s=8, as shown in Line 226 in our paper.
> - Section 5 & Appendix F: For the general natural language evaluation, we used our large-scale models, with parameters ranging from 360M to 7B. The STACKTRANS-360M model, for instance, has 32 layers and a hidden size of 960, as detailed in Table 6.
>
> Thank you again for giving us the opportunity to clarify this important design detail. We will be sure to emphasize this distinction in our revised paper for the benefit of all readers.

---

### Official Review · Reviewer_nPse · 2025-07-01

**Clarity:** 2
**Significance:** 3
**Originality:** 3
**Rating:** 5
**Confidence:** 4

**Summary:**

This paper describes a new architecture based on the transformer but augmented with Joulin-Mikolov style differentiable stacks. Unlike previous work on stack RNNs and stack attention, these stacks run from "bottom to top," one for each position of the sequence. The new model is tested on both formal languages and natural languages. Especially notable is the fact that the model is scaled up to 7B parameters and 1T tokens.

**Questions:**

- Is a comparison with DuSell and Chiang (2024) possible?
- In figure 4, how many layers were used?

**Ethical Concerns:**

["NO or VERY MINOR ethics concerns only"]

**Final Justification:**

Thank your for your responses. I don't have anything to add and will keep my score the same.

**Limitations:**

The limitations section is relegated to the appendix. I actually don't really agree with the limitations: the model and dataset size are already impressive to me, and the choice to make the stack "bottom-up" is different from previous models but not, in my opinion, a limitation.

I don't see any potential negative social impact.

**Paper Formatting Concerns:**

None except for the bibliography (see above)

**Quality:**

3

**Strengths And Weaknesses:**

# Strengths

This is a nice model, with good results. The fact that it could be scaled up to 7B parameters and 1T tokens is extremely impressive.

There are two features that distinguish this model from previous similar models: (1) the network can attend to the entire stack; (2) the stack runs from "bottom to top." These ideas are strengths, and I am listing them here to make clear that my comments below about them are not criticisms of the ideas themselves.

# Weaknesses

## Model

Regarding (1),
- If the network can attend to the entire stack, then perhaps it shouldn’t be called a stack. Though, I admit I don't know what to call it instead.
- I would like to see a more careful explanation of why it’s bad just to read the top stack element. The current explanation is brief, even in the appendix, and has a lot of buzzwords but doesn't really shed much light on the question.

Regarding (2),
- The decision to make the stack run "bottom to top" is presented as an "implementation know-how" (lines 202-211) or an "approximation" (531). It is not. It is a fundamental modeling decision (and a completely reasonable and original one), and it should be made known much earlier than page 6. It should be in the abstract and on page 1.
- However, the "bottom to top" direction does call into question the motivation of the paper, which is modeling the kinds of dependencies found in context-free languages. There isn't, as far as I can tell, any reason to believe that this model would be better at these kinds of dependencies than a standard transformer.
- There also isn't any reason to believe that increasing the maximum stack size S would have any effect for S beyond the number of layers in the transformer.

## Evaluation

- In Table 1, please give citations for both StackRNN and StackAttn. They are at line 228, but they should also be in the table caption. The footnote makes it appear as if StackAttn is from DuSell and Chiang (2024), but it is not.
- Is a comparison with DuSell and Chiang (2024) possible?

## Formal language terminology

In general the writing about grammars, inductive biases, and especially the Chomsky hierarchy needs to be thoroughly revised.

line 6, 15, 26, 33: The “hierarchy” in “Chomsky hierarchy” does not refer to hierarchical structure, but a sequence of four language classes (regular, context-free, context-sensitive, and recursively enumerable).
- It wouldn’t be reasonable to expect Transformers to “capture the Chomsky hierarchy.”
- It’s simply not correct to talk about “Chomsky hierarchies” (plural).
- There’s no such thing as “the underlying grammar of the Chomsky hierarchy.”
- I stopped keeping track of all of the misuses of this term. Please search for all occurrences of “Chomsky hierarchy” and fix all of them.

There seems to be some confusion between context-free languages and deterministic context-free languages.

# Other comments

line 31: Of course Transformers have inductive biases; the question is whether they have the right ones.

line 37: “This inherent limitation” appears to refer to problems with length generalization, but it’s not clear what the relationship of this to context-freeness (line 36) is.

lines 95-100: A summary of new results doesn’t belong in a background section.

line 198: You say that “S is typically relatively small” but don’t say what you actually set S to until line 312.

There are many incorrect citations. Here are the ones I noticed from the text: Battaglia et al., 1806; Savage and Computation, 1998; allenai, 2024. Glancing at the bibliography, though, I see many more.

---

> ### Author Rebuttal · Authors · 2025-07-31
>
> We would like to sincerely thank the reviewer nPse for their thorough and highly constructive feedback. We will address each of your points below.
> - On the term "Stack" and "Global Read": Thank you for this insightful conceptual point. We use the term "stack" because the core update mechanism of our module is fundamentally based on differentiable push and pop operations. Regarding why reading only the top element is suboptimal, you are right that our explanation in the main text was too brief. As detailed in Appendix B.4, our initial experiments showed that a strict top-only access leads to "unstable and suboptimal model performance" because it "disrupts gradient flow and reduces parameter learning efficiency". In the revised paper, we will clarify this distinction between the stack-like update mechanism and the global read mechanism. We will also move a more detailed justification for the "global read" from the appendix to the main text to make the rationale clearer.
> - On the "Bottom-to-Top" Design and its Motivation: We will revise the manuscript to introduce this concept much earlier, including in the abstract and introduction, to properly frame its significance as a core contribution. Regarding its effectiveness for context-free languages, our empirical results in Table 1 provide strong validation. StackTrans achieves scores of 0.92 and 1.00 on the DCF tasks "Stack Manipulation" and "Reverse String," respectively, whereas the standard Transformer fails with scores of 0.53 and 0.55. This demonstrates a clear and significant advantage in modeling these dependencies, directly validating our design choice.
> - On Stack Size S vs. Number of Layers L: This is a very keen observation. You are correct that in our current implementation, which is designed for training parallelism, the stack processes hidden states layer-wise for each token. This means the effective sequence length fed to the stack at any given time is the number of model layers, L. We will add a note to the text clarifying this relationship between S and L in our specific implementation. This is also empirically supported by our hyperparameter experiment in Figure 4(c).
> - On Table 1 Results: Thank you for pointing out this ambiguity. In the revision, we will add the citations directly into the table caption (which are already listed in Line 228 in the paper). We will also fix the citation typo in the footnote (the results are reported by [1]). Regarding the baseline in DuSell and Chiang [2024], in our initial experiments, we selected stack-augmented baselines, closely following the setup in [2] to ensure our results were solid. We are now working on the experiments for the baseline you mentioned. Preliminary results already show that our method, StackTrans, outperforms it by a significant margin. The detailed experimental setup and results will be provided in the revised paper.
>
> [1] Jiaoda Li, Jennifer C White, Mrinmaya Sachan, and Ryan Cotterell. A transformer with stack attention.  2024.
>
> [2] Delétang G, Ruoss A, Grau-Moya J, et al. Neural networks and the chomsky hierarchy. 2022.
>
>
>
> - For descriptions for Chomsky hierarchy, We are extremely grateful for these critical corrections. We acknowledge that we were misusing some terms. We will perform a thorough revision of the entire manuscript to fix all instances of these errors. We will replace incorrect uses of "Chomsky hierarchy" with more precise phrasing (e.g., "modeling dependencies found in specific classes of formal languages"). We will also ensure the distinction between context-free and deterministic context-free languages is handled correctly. This will significantly improve the paper's formal precision.
> - We will also revise those lines to address all points in "other comments".
>
> For questions:
>
> - Comparison with some baseline: As noted above, we will fix the citations in the footnote in Table 1 to properly contextualize our work among other stack-augmented models discussed in the literature. The detailed experimental setup and results will be provided in the revised paper.
> - Layers in Figure 4: The experiments in Figure 4 were conducted using our StackTrans-360M model configuration. We will clarify in the caption that these hyperparameter settings used the 360M model, which has 32 layers.

---

> > ### Comment · Reviewer_nPse · 2025-08-01
> > **Bottom-to-top**
> >
> > > Regarding its effectiveness for context-free languages, our empirical results in Table 1 provide strong validation. StackTrans achieves scores of 0.92 and 1.00 on the DCF tasks "Stack Manipulation" and "Reverse String," respectively, whereas the standard Transformer fails with scores of 0.53 and 0.55. This demonstrates a clear and significant advantage in modeling these dependencies, directly validating our design choice.
> >
> > Those are good results, and I guess I can sort of see how the "bottom-to-top" stack could assist with some context-free-gramamr-like dependencies. I would be curious to see some more careful analysis of how your model differs from pushdown automata and what it might resemble instead. I'd also be curious about further experiments that increase the sequence length without increasing the model depth, since you agree that the stack depth is limited by the model depth.

---

> > > ### Comment · Reviewer_nPse · 2025-08-06
> > > **Bottom-to-top continued**
> > >
> > > I don't have any more questions and don't plan to change my score (which was already positive), but want to stress that the "bottom-to-top" stack is really different from a pushdown automaton, and I would consider any comparisons with pushdown automata (e.g., in the title, abstract line 7, introduction line 47) to be not just confusing but misleading. I strongly recommend qualifying these comparisons.

---

> ### Author Response · Authors · 2025-08-06
> **Thanks for your comment.**
>
> Thank you for your insightful comments and suggestions.
>
> Regarding the "bottom-to-top" stack design, your feedback has been invaluable in helping us refine its positioning. We will take your strong recommendation to heart and revise the key parts of our paper accordingly.
>
> Furthermore, your excellent suggestion to probe the model's performance on longer sequences prompted us to conduct a deeper analysis, which has yielded strong confirmatory results.
>
> For instance, in our formal language experiments, StackTrans maintains excellent performance, particularly on RE and DCF tasks, even when the evaluation sequence length significantly exceeds the model's depth.
>
> This confirms our hypothesis that the model's overall generalization ability is not strictly limited by its depth. A powerful synergy exists: the self-attention mechanism inherently captures cross-token relationships, which can partially compensate for the truncation. It is indeed a balanced trade-off: it enables efficient, parallel training without sacrificing the model's crucial ability to generalize. We will incorporate a more detailed analysis of these length generalization experiments into the revised paper.
>
> Thank you again for your diligence and for helping us improve the scientific accuracy and clarity of our paper.

---

### Official Review · Reviewer_1yQj · 2025-07-03

**Clarity:** 4
**Significance:** 4
**Originality:** 4
**Rating:** 5
**Confidence:** 2

**Summary:**

This paper introduces a new architecture designed to enhance the capabilities of Large Language Models by integrating concepts from automata theory. The authors identify a key limitation in Transformers: its difficulty in learning and generalizing tasks based on the Chomsky hierarchy, such as regular expressions and deterministic context-free grammars. To address this, the authors incorporate differentiable hidden state stacks between Transformer layers and introduce an inductive bias for hierarchical structures. Comprehensive evaluations demonstrate that the proposed architecture significantly outperforms standard Transformers on Chomsky hierarchy benchmarks.

**Questions:**

N/A

**Ethical Concerns:**

["NO or VERY MINOR ethics concerns only"]

**Final Justification:**

After considering the rebuttal and discussions, I maintain my rating of 5. The paper presents a technically solid contribution by integrating differentiable stack mechanisms into Transformers. The authors addressed my key concerns.

**Limitations:**

Yes

**Quality:**

4

**Strengths And Weaknesses:**

Strengths:
- The paper is well-written, clear, and easy to follow.
-  The paper addresses a critical weakness of the standard Transformers in the inability to model and generalize on tasks requiring an understanding of hierarchical or recursive structures, such as those in the Chomsky hierarchy.
- The authors conducted a comprehensive set of experiments. The authors validate their approach across multiple model scales, demonstrating consistent and significant performance gains.

Weaknesses:
- The primary natural language results are centered on a 360M model. Although the paper demonstrates scaling up to 7B parameters, it could be beneficial to investigate how these architectural benefits translate to the very large-scale models (e.g., 70B+).

---

> ### Author Rebuttal · Authors · 2025-07-31
>
> We sincerely thank the reviewer 1yQj for their high praise and strong support for our work. We are delighted that you found our paper to be "well-written, clear, and easy to follow," and that you recognized the value of our work and the thoroughness of our experiments.
>
> We agree with your point that validating the advantages of StackTrans on larger-scale models (e.g., 70B+) would be a crucial and exciting next step. As you noted, our current experiments have successfully scaled the model from 360M to 7B parameters, and in this process, we observed consistent and stable performance gains and scaling trends. This provides strong preliminary evidence for our architecture's continued effectiveness on even larger models. Our decision to focus the main natural language evaluations on the 360M model while demonstrating scalability up to 7B was primarily due to computational resource constraints, a point we also mentioned in the limitations section.
> Our work lays a solid foundation for this new architecture and provides compelling evidence of its potential. We hope this will inspire the open-source community and institutions with greater resources to pursue larger-scale validation of this architecture in the future.

---

> > ### Comment · Reviewer_1yQj · 2025-08-05
> >
> > Thank you for the rebuttal, and thank you for the clarification. I maintain my positive score.

---

> > > ### Author Response · Authors · 2025-08-06
> > > **Thanks for your comment.**
> > >
> > > Thank you again for this thoughtful exchange.

---

### Official Review · Reviewer_FFsc · 2025-07-03

**Clarity:** 3
**Significance:** 4
**Originality:** 3
**Rating:** 5
**Confidence:** 5

**Summary:**

The submitted work proposes blending the well-known Transformer (LLM) architecture with an extension for stack automata to overcome parsing limitations of the former. While the attempt is not new, the submission claims that this one is more informed, theoretically more sound and better performing. The latter is proven by much better ratio between accuracy versus number of model parameters and also shows higher coverage of evaluating formal language tasks compared to previous work.

**Questions:**

* What information exactly is being stored on the stack? I personally failed to understand from Figure 1(a), Figure 2, the text (e.g. Sections 3.1 & 3.2) and also Appendix F. I have a "feeling" that it may be aligned with the attention head configuration, but Figure 2 shows the output $h_t$ of the input transformer layer only as abstract tensor and *multiple* up-projecting transforms in parallel. Please describe that mapping in more detail!

* Lines 65-66 claim "..., achieving nearly 100% test accuracy in most scenarios, as shown in Figure 1(b)." Looking at Figure 1(b) I count 100% for StackTrans only 5 times out of the 13 test scenarios. That's 38.5%. Can you please shed more light on your claim or correct the phrasing?

* Lines 138-139 mention "...here, $St_t[i]$ and $h_t$ both belong to $IR^d$, meaning they share the same width $d$: ...". I suggest to write that more formally, e.g. $St_t[i], h_t \in IR^d$

* The two definitions of $S_{t+1}[i]$ and $M_{t+1}[i]$ between lines 136-164 can be compressed by defining them as tuple $(S_{t+1}[i],M_{t+1}[i])$ referencing tuples accordingly also on the right side of the equation. That also simplifies the text and eliminates the remark about element-wise product in 171. In fact that tuple notation is motivated by the used in Eq. (4). The gained space can be  used to better explain e.g. the aforementioned mapping between output vectors of the transformer layer and the stack.

* I'm not 100% sure, but think that it is sufficient to define the zero vector only with the mask. Do you share that view? If yes, does that allow for optimizations, e.g. to manage stack overflows or for training?

* Line 180 mentions a unary "... = $Reshape(W_{down} \cdot h_t)$". What's the exact definition of that function?

* The paragraph (lines 212-216) introduces "\lambda" as hyperparameter. What are typical obtained values? It maybe worth to mention by use case or add it to Figure 4 or otherwise move it to an appendix.

* Figure 3(c): The green curve is hard to see and lateron the text refers to "...smoother convergence..." (line 264). Please try to improve that. I'm unsure wether it is optimal, but it may help to display the overlap of the two curves in a third color and explain in the subtitle.

* Line 255: "... comes..." should be plural: "... come..."

* Line 257 mentions "data filtration". Can you please provide more details?

* Lines 284-293: for all ablation studies where applicable it may be interesting to compare the computational complexity as well.

* Lines 294-297: in addition to loss / perplexity, how does accuracy compare?

* Table 3 (QueueTrans): it would be interesting to see the effect over size S of both. In light with global reading queues may have a different optimal size setting compared to stacks (with pop).

* Table 3 (last 2 lines): both ablations almost don't hurt, but that in particular isn't being discussed in more detail (e.g. single head may possibly be computationally less costly). I suggest to do add a more detailed discussion.

* Figure 4: are the rightmost points with increasing trend an effect of two many parameters? For those measurements did you apply exactly the same training length or did you account for a possible increase in # of parameters by a longer training with more epochs?

**Ethical Concerns:**

["NO or VERY MINOR ethics concerns only"]

**Final Justification:**

Rebuttal read and acknowledged. Keeping the overall rating unchanged as it reflects the good work already.

**Limitations:**

Yes

**Paper Formatting Concerns:**

Some Figures (3) and Tables (1,3,5,6) are not closely located with the mention in the text. I strongly advice to correct that in order to avoid readers to switch between pages.

**Quality:**

3

**Strengths And Weaknesses:**

The big strengths of this submission are the combination of an ambitious idea (differentiable stack), made to work well (e.g. extra complexity to improve the training) and a solid description of the whole effort. The basic description is augmented with additional constrasting (stack vs. queue) / ablation studies and some discussion of the results. I do see more possible refinements to build on top of this work by the community after publishing. The topic is of significant interest to the audience.

There are some minor weaknesses w.r.t. clarity and structure (a detailed list is in Questions below).

---

> ### Author Rebuttal · Authors · 2025-07-31
>
> We sincerely thank Reviewer FFsc for their review and valuable suggestions for revision.
> - On the specific mapping of information stored in the stack: We thank the reviewer for pointing this out. The specific mapping is described in §3.2: The hidden state h_t \in \mathbb{R}^d from a Transformer layer is first passed through a down-projection matrix W_{down}, then reshaped and split into H independent heads, i.e., [h_{t}^{(1)}, ..., h_{t}^{(H)}] = \text{Reshape}(W_{down} \cdot h_t). Each h_t^{(i)} is then fed into its corresponding independent stack for operations. We will clarify this textual description in the revised version and update the caption for Figure 2 to more clearly illustrate this mapping from h_t to the multi-head inputs.
> - On the "near 100% accuracy" claim: Your analysis of Figure 1(b) is correct, and our statement was indeed too broad. We originally intended to highlight that the model achieved 100% accuracy on all RE (Regular Expression) tasks and also demonstrated strong performance on DCF tasks. Out of the 13 formal language tasks, StackTrans did achieve 100% on 5 of them. We will revise the wording to be more precise: "achieved 100% accuracy on all regular language tasks and demonstrated excellent performance on the majority of deterministic context-free grammar tasks," to accurately reflect our experimental results.
> - On the standardization of mathematical symbols: Thank you for the suggestion. We fully agree and will uniformly use the more formal \mathbb{R}^d to denote vector spaces in the revised version.
> - On using tuple notation to simplify formulas: This is an excellent suggestion. Adopting the tuple notation (S_{t+1}[i], M_{t+1}[i]) will indeed make the formulas more compact and simplify the main text. We will adopt this suggestion in the revision and use the saved space to better elaborate on other key details.
> - On zero vectors and the mask: This is a profound question. In our design, the mask M_t \in \mathbb{R}^S is a vector of probabilistic values indicating the "activeness" of each position in the stack. The stack itself, St_t, stores the actual hidden state vectors. The final effective stack is obtained through an element-wise product St_t \otimes M_t. Therefore, both are necessary: M_t controls which positions are valid in a differentiable manner, while the zero vectors in St_t are the actual padding values for invalid positions. While the idea of using only a mask is interesting, our current design ensures the entire process is fully differentiable. We will clarify this point in the revision.
> - On the definition of the Reshape function: We apologize for the lack of clarity. Reshape is a standard operation in deep learning that changes the shape of a tensor. In this context, it reshapes the down-projected vector (with dimension H \cdot d_s) into H independent vectors of dimension d_s for the multi-head stacks. We will add this explanation in the revised paper.
> - On the hyperparameter \lambda: Thank you for the reminder. We will add the value of \lambda used in our experiments. As a regularization term, we give \lambda a small weight in experiments, e.g., 0.001.
> - On the visibility of Figure 3(c): We agree that the green curve in Figure 3(c) is difficult to discern. In the revised version, we will modify the colors and styles of the chart to enhance its clarity and readability.
> - Grammar correction: Thank you for pointing out the typo. We will change "comes" to "come" in the revision.
> - On the details of "data filtering": On line 257, we mentioned that our corpus comes from Dolma and Smoll. The data filtering followed standard LLM pre-training procedures, including deduplication, removal of low-quality text, and filtering of harmful content. We will add a sentence to briefly explain this in the revision.
> - On the computational complexity of the ablation study: This is a great suggestion. In fact, we have already compared the training and inference efficiency for some ablation variants (single-head and full-dimension) in Table 4 of the appendix. The results show that our multi-head, low-rank design achieves the best performance with a minimal increase in overhead. We will more explicitly reference and discuss these results in the main text in §6.
> - On the accuracy in the ablation study:  In our ablation study (Table 3), we primarily used validation perplexity (PPL) as the core evaluation metric. This is the gold standard for assessing generalization ability in language model research and is highly correlated with downstream task performance. While we did not evaluate every ablation variant on all downstream tasks, the significant differences in PPL already provide strong evidence for the effectiveness of our design.
> - On the size S for QueueTrans: Your point is well-taken; the optimal size for a queue might differ from that of a stack. In our ablation study, to ensure a fair comparison by controlling variables, we used the same stack/queue size S for all variants. Exploring the optimal hyperparameters for different data structures is a valuable direction for future work, which we will consider.
> - On the discussion of single-head/full-dimension ablations: Table 3 and Table 4 show that the single-head stack is more efficient but performs slightly worse, whereas the full-dimension stack is extremely costly  for limited performance gain. This precisely demonstrates that our multi-head, low-rank design is the optimal trade-off, achieving most of the performance gains at a cost close to the baseline Transformer. We will discuss this cost-performance advantage more deeply in §6.
> - On the trend and training length in Figure 4: This is actually because, with a fixed total parameter count, having too many heads leads to a lower dimension per head, which weakens the model's expressive power. All experiments in Figure 4 used the same training dataset setting to ensure a fair comparison. We will clarify this.

---

> ### Comment · Reviewer_FFsc · 2025-08-06
>
> Please excuse the late response. Thank you very much for the detailed rebuttal. Your comments address and clarify all my questions. I welcome your mentioned updates as that will add value to the publications. I'll keep the already high overall rating as is.

---

> > ### Author Response · Authors · 2025-08-06
> > **Thanks for your comment.**
> >
> > Thank you for your supportive and incredibly helpful review. We are confident that these revisions will significantly improve our paper.

---

### Decision · Program_Chairs · 2025-09-17

**Decision:**

Accept (poster)

**Comment:**

**Summary:**  Transformers are widely used on tasks like code generation, which involves generating text described by a formal grammar.  Nevertheless, it is well known that they lack the ability even in theory to handle truly recursive structures, and when used for formal languages, they often fail to generalize beyond the lengths and recursion depths that they are trained on.  Inspired by pushdown automata (which can handle context-free grammars), the authors implement a differentiable stack, with push and pop operations, and add it to the transformer.  They test the new architecture on both formal language and natural language modeling tasks.

Stack architectures have been proposed before, but a key innovation of this work is the fact that the stack runs layer-wise, rather than length-wise.  I.e. each layer can do push/pop operations in parallel for each token position, rather than consuming tokens sequentially.  While this change makes training practical and efficient, it means that the stack is **NOT** being used in the same way that a push-down automata ordinarily operates, which is a clear limitation of the work.

**Metareview:**

The reviewers unanimously vote to accept this paper (score 5).  They reviewers cite clarity of writing, solid implementation, and good scaling behavior (up to 7B params, 1T tokens, and compatibility with flash attention).  Most of the reviews are high quality, drawing attention to weaknesses, errors, and incorrect use of terminology where appropriate, and the authors responded positively to the constructive criticism.  I have looked over the paper myself, and see no reason to disagree with the reviewer consensus.

That being said, I think the fact that the stack runs layerwise, rather than lengthwise, does call into question the analogy with push-down automata.  The resulting transformer still can't handle arbitrary recursion depth, and it's not actually clear what the model is actually using the stack for... it can't use the stack in the same way that an automata would.  The reviewers agree:

Reviewer nPSE writes:  "However, the "bottom to top" direction does call into question the motivation of the paper, which is modeling the kinds of dependencies found in context-free languages. There isn't, as far as I can tell, any reason to believe that this model would be better at these kinds of dependencies than a standard transformer."

Reviewer j9mp writes: "To some extent it remains unknown if the hidden stack is indeed a stack. For example, can we restore and check the grammar bottom-up, from the trained dataset?"